# Beyond Nash Equilibrium: Bounded Rationality of LLMs and Humans in Strategic Decision-making

## Abstract

Large language models are increasingly used in strategic decision-making settings, yet evidence shows that, like humans, they often deviate from full rationality. In this study, we compare LLMs and humans using experimental paradigms directly adopted from behavioral game-theory research. To ensure a precise comparison, we experiment with a finite action space, a closed-form Nash equilibrium benchmark, and well-documented human deviations from that equilibrium. These criteria naturally yield two complementary game families: a zero-sum family exemplified by Rock-Paper-Scissors and a non-zero-sum family exemplified by the Prisoner's Dilemma, whose parameters we systematically vary to form a broader task space. By placing LLMs in identical experimental conditions, we evaluate whether their behaviors exhibit the bounded rationality characteristic of humans. Our findings show that LLMs reproduce similar human heuristics, such as outcome-based strategy switching and increased cooperation when future interaction is possible, but LLMs follow these patterns more rigidly and demonstrate weaker sensitivity to the dynamic changes in the game environment. Model-level analyses reveal distinctive architectural signatures in strategic behavior, yet reasoning-enhanced models still falter in dynamic, opponent-adaptive settings, showing that a long chain of thought alone does not guarantee full rationality. These results indicate that current LLMs capture only a partial form of human-like bounded rationality and highlight the need for training methods that encourage flexible opponent modeling and stronger context awareness.

## 1 Introduction

Strategic decision-making is the process of identifying optimal actions to achieve specific goals. This involves calculating optimal solutions based on rules defining the task, predicting the behavior of others from past interactions, and adapting to an evolving environment. (Wikipedia contributors, 2025). As large language models (LLMs) have grown in capability, recent research has begun to investigate their potential as rational agents in strategic settings (Hua et al., 2024; Jia et al., 2025). However, a consistent finding from empirical studies using game-theoretic frameworks is that LLMs often diverge from theoretically optimal Nash-equilibrium strategies (Akata et al., 2025; Hayes et al., 2024). This sub-optimality is attributed to several factors, including systematic cognitive biases, misunderstanding of task specifications, and challenges in updating beliefs or predicting others' actions in evolving environments (Fontana et al., 2024; Hayes et al., 2024).

This deviation from rational behavior is not unique to LLMs. Human strategic decision-making is also known as "bounded rationality", which describes how cognitive limitations, perceptual filters, and contextual factors cause decisions to diverge from optimal standards (Kahneman & Tversky, 1979; Tversky & Kahneman, 1974). Since modern LLMs are learnt from vast amounts of human-generated data, a critical question emerges: do the patterns of bounded rationality observed in LLMs mirror those of humans? (Guo et al., 2024; Cheung et al., 2025). The similarity between them reflects LLMs' internalization of strategic reasoning akin to human cognition. Understanding this connection is vital for guiding the development of LLMs that can adapt more fluidly to real-world strategic interactions and achieve better optimal outcomes (Guo et al., 2024; Lyu et al., 2025). Therefore, a systematic investigation into **the bounded rationality between LLMs and humans**

**in strategic decision-making** is of critical value, yet existing research has not adequately addressed this area.

To investigate this question empirically, it is necessary to design a strategic environment that can precisely measure and compare the behaviors of both LLMs and humans, satisfying three core criteria: (i) **a finite action space** to allow for a precise quantitative analysis of strategic behaviors; (ii) **a clear, closed-form Nash equilibrium** to serve as a normative benchmark for rational behaviors; and (iii) **an extensive behavioral literature documenting how humans departure from equilibrium play**, which provides a robust empirical baseline for comparison. These requirements lead us to two classic strategic games: the zero-sum contest of **Rock-Paper-Scissors (RPS)** and the non-zero-sum **Prisoner's Dilemma (PD)**. In RPS, the unique equilibrium is a mixed strategy of selecting each action with equal probability, while in PD, the Nash equilibrium of mutual defection starkly contrasts with the Pareto-optimal outcome of mutual cooperation. Their complementary decision structures provide us a robust framework to not only assess whether LLMs approximate optimal strategies but also to examine whether their failures replicate the predictable, bias-driven patterns characteristic of human strategic decision-making (Wang et al., 2014; Hoffman et al., 2015; Zhang et al., 2021; Ahn et al., 2001; Schneider & Shields, 2022).

We replicate the settings human-subject experiments of RPS and PD (Zhang et al., 2021; Bó, 2005) for LLMs: placing each LLM in the same decision-making environments. After logging their action sequences, on the one hand, we compute aggregate performance metrics, such as choice percentages in RPS and cooperation rates in PD, to benchmark against human behaviors. On the other hand, we analyze each LLM individually to understand how factors like model family and reasoning mechanisms influence LLMs' strategic behavior. This two-stage approach provides a comprehensive view of how LLMs align with or diverge from human strategic decision-making.

Our experiments reveal three key findings. **First**, LLMs adopt human-like decision shortcuts, but in a much more rigid manner. **Second**, LLMs are less adaptive when responding to environmental changes than humans. **Third**, strategic behavior is consistent within model families, creating distinct "strategic signatures". Notably, reasoning-enhanced models (such as DeepSeek-R1) can solve for the Nash equilibrium in analytically clear games but struggle in scenarios that demand theory-of-mind like inference (Premack & Woodruff, 1978).

Our study introduces a general recipe for direct, head-to-head comparisons between human and machine strategic behaviors by placing LLMs into human-subject experiments. We apply the framework to a suite of leading LLMs, matching them against human participants in repeated rounds of two representative games to analyze their bounded rationality and strategic heuristics. The findings underscore the need for training methods focused on enhancing opponent modeling and context-aware reasoning to advance LLMs toward human-level strategic adaptability.

## 2 RELATED WORK

**Humans' Strategic Behaviors** Evidence from behavioral game theory shows that humans consistently diverge from Nash equilibrium. Limited iterative thinking explains non-equilibrium choices in guessing and matrix games (Nagel, 1995; Stahl & Wilson, 1995). Social preferences lead players to sacrifice payoffs for fairness or reciprocity (Rabin, 1993; Fehr & Schmidt, 1999; Charness & Rabin, 2002). In repeated interactions, costly penalty and intuitive cooperation keep contributions above equilibrium levels (Fehr & Gächter, 2000; Rand et al., 2012). Together these results confirm bounded rationality as a regular feature of human play (Camerer, 1997).

Classic examples of bounded rationality appear in games like Rock-Paper-Scissors (RPS) and Prisoner's Dilemma (PD). In RPS, human players tend to repeat a move after a victory and switch after a defeat, producing cyclic patterns rather than truly random choices (Dyson et al., 2016; Wang et al., 2014). In iterated PD, cooperation often exceeds the one shot equilibrium prediction: participants use tit for tat, punish defectors and cooperate more when future interaction is likely (Bó, 2005; Montero-Porras et al., 2022; Rand et al., 2012). These game-specific tendencies further underscore bounded rationality in human strategic play.

**LLM Performance in Strategic Decision-making Tasks** A rapidly expanding body of work evaluates LLMs in a spectrum of strategic contexts, from simple 2×2 matrix games to rich multi-agent

environments. Existing works have tested models in classic two-player games, coordination games, public-goods dilemmas, negotiation and bargaining tasks, large-scale board games (e.g., Diplomacy), and social-deduction settings (e.g., Werewolf) (Lorè & Heydari, 2024; Gandhi et al., 2023; , FAIR; Hua et al., 2024; Ye et al., 2025; Mozikov et al., 2024; Xia et al., 2024; Kwon et al., 2024; Deng et al., 2024; Bianchi et al., 2024; Fish et al., 2025). This literature shows that while advanced LLMs can plan, cooperate, and even outperform humans under certain conditions, their strategic behavior remains highly sensitive to model family, prompting style, and auxiliary planning modules.

Simple strategic environments with clear Nash equilibrium provide a rigorous test of LLM rationality. In the Ultimatum Game, GPT-4 rejects low but positive offers more often than predicted, accepting only larger splits (Tennant et al., 2025; Kwon et al., 2024). In Matching Pennies, models fail to randomize uniformly, showing stable side biases absent explicit randomization prompts (Haag & Kruse, 2024). Reasoning-tuned variants such as DeepSeek-R1 and o1 approach equilibrium play more closely in one-shot dilemmas, yet still rely on chain-of-thought scaffolding to achieve optimal strategies (Brookins & DeBacker, 2023; Silva, 2024), showing that even advanced LLMs exhibit bounded rationality without targeted prompting.

**Comparing Human and LLM Behavior in Strategic Decision-making**   Recent studies compare large language models with humans in three main ways. First, large scale replications give models exactly the same behavioral economics tasks used with laboratory subjects and then match the resulting choice distributions (Mei et al., 2024; Zhou et al., 2025; Yu et al., 2025; Herr et al., 2024). Second, head-to-head designs pair model agents with live human players in repeated games and record relative payoffs and learning curves (Akata et al., 2025). Third, simulation benchmarks check model outputs against documented human heuristics and bias patterns (Fontana et al., 2024; Lorè & Heydari, 2024; Hosseini & Khanna, 2025; Hagendorff et al., 2023; Macmillan-Scott & Musolesi, 2024). Taken together, these work shows that current models often match human averages on cooperation and fairness but still diverge in coordination skill and in how bounded rationality manifests.

However, existing studies rarely replicate full sociological experiment protocols, limiting direct comparisons under identical conditions. We fill this gap by implementing parallel sociological experiments for both human participants and LLMs, allowing a one-to-one assessment of their strategic decision-making. Moreover, by focusing explicitly on bias and bounded rationality in classic games, we examine whether models and humans exhibit similar systematic deviations from optimal play, thus uncovering the shared and distinct mechanisms underlying their decisions.

## 3   METHODOLOGY

To examine whether LLMs exhibit human-like or fully rational patterns in strategic decision-making, we conduct simulation-based experiments modeled after established human-subject protocols.

**Model Selection**   We evaluated six state-of-the-art LLMs from three distinct model families: GPT-4o-2024-11-20 (GPT-4o) and o1-2024-12-17 (o1) (OpenAI, 2024a;b), Claude-3.5-sonnet-2024-10-22 (Claude-3.5) and Claude-3.7-sonnet-2025-02-19 (Claude-3.7) (with extended thinking) (Anthropic, 2024; 2025), and DeepSeek-V3, DeepSeek-R1 (DeepSeek-AI, 2025b;a). These models are selected to represent a range of reasoning capabilities and alignment strategies, including both general-purpose dialogue agents and models fine-tuned for structured multi-step inference. This diversity allows us to assess how strategic behavior varies across model architectures and training regimes.

**Experiment Design and Prompting**   We replicate **identical** experiment conditions from the behavioral game theory literature that study human strategic decision-making (Zhang et al., 2021; Bó, 2005), adapting these settings for LLMs to enable credible comparisons. When constructing prompts, we closely align the language, incentives, feedback structure, and instructions with those used in human-subject experiments, ensuring consistency in framing and strategic context. For example, the payoff matrices of Table 1(a) and Table 1(b) are directly taken from the original human-subject experiments (Zhang et al., 2021; Bó, 2005).

Table 1: Two payoff matrices for both games. Entries shown as (Player 1, Player 2). Units in points.

(a) Rock–Paper–Scissors

| Player 2 | Player 1 | | |
|---|---|---|---|
| | Rock | Paper | Scissors |
| Rock | (2, 2) | (1, 3) | (4, 0) |
| Paper | (3, 1) | (2, 2) | (1, 3) |
| Scissors | (0, 4) | (3, 1) | (2, 2) |

(b) Prisoner's Dilemma

| Player 2 | Player 1 | |
|---|---|---|
| | Cooperate | Defect |
| Cooperate | (65 , 65) | (100 , 10) |
| Defect | (10 , 100) | (35 , 35) |

Each model is queried with the same prompt format that includes the current game context, payoff structure, and recent action history. The model is instructed to output both an action (e.g., "Rock" or "Cooperate") and an explanation of its choice. The explanation serves to elicit the model's reasoning process in a Chain-of-Thought style, enabling even general-purpose models to demonstrate a degree of strategic reasoning and making their decision logic more interpretable.

**Comparison and Evaluation** We evaluate LLM behaviors using a unified set of metrics grounded in game-theoretic principles and established behavioral benchmarks (Zhang et al., 2021; Bó, 2005). Based on existing human studies, we adopt relevant evaluation metrics such as choice distribution in Rock-Paper-Scissors and cooperation rates in the Prisoner's Dilemma. These metrics were originally designed to analyze human decision-making and to reveal characteristics of bounded rationality. By applying the same metrics to LLM outputs, we assess the extent to which model behavior aligns with or deviates from human behavior under comparable experiment settings. All human behavioral data used for comparison are sourced **directly from the original studies** (Zhang et al., 2021; Bó, 2005).

In addition, we perform model-level analyses to examine behavioral variability across different model architectures, providing insights into how model design influences strategic reasoning. To complement these quantitative results, we further analyze selected examples of model reasoning for a better understanding of the cognitive patterns, decision heuristics, and potential limitations underlying model behavior.

## 4 EXPERIMENTS

In this section, we first present the experiment setups and results obtained under the two competitive games, and summarize the findings from the two games.

### 4.1 ROCK-PAPER-SCISSORS

Rock-Paper-Scissors (RPS) offers a simple yet powerful framework to study competitive decision-making in adversarial settings. Since no option strictly dominates the others, the game has no pure-strategy equilibrium. Instead, the optimal strategy is to play randomly, choosing each option with an equal probability.

Despite the theoretical predictions, empirical research has shown that human participants struggle to approach true equilibirum. Instead, players often rely on predictable behavioral patterns such as frequency bias or sequential strategies based on past outcomes. One such strategy is Win-Stay/Lose-Change (WSLC), where players repeat a choice after a win and switch after a loss. (Wang et al., 2014; Hoffman et al., 2015).

Building on these insights, we investigate whether LLMs exhibit similar behavioral patterns when placed in sequential adversarial settings. In particular, we explore whether LLMs generate action sequences that deviate from Nash equilibrium. This approach allows us to assess whether LLMs internalize basic principles of adaptive play and whether their decision-making resembles that of human participants.

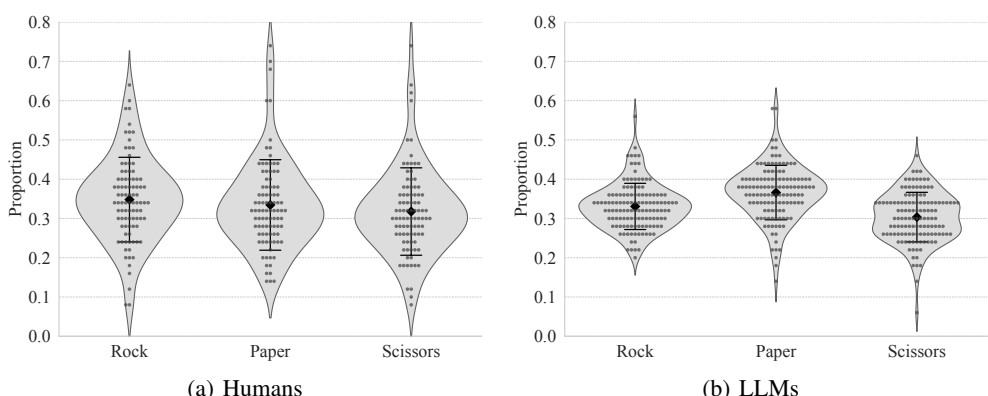

Figure 1: The proportion of choices selected by (a) humans and (b) LLMs.

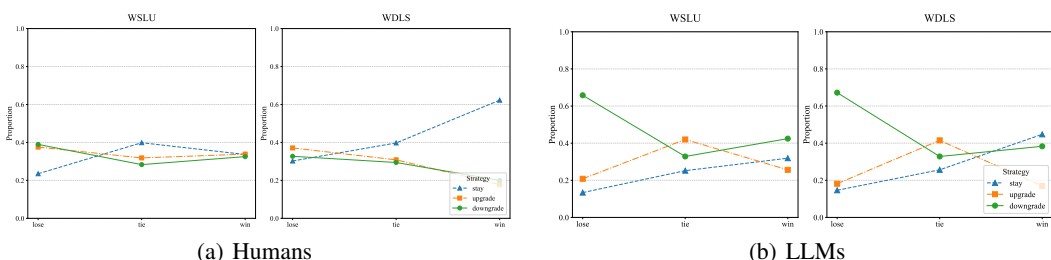

Figure 2: Comparison of outcome-conditioned strategy distributions under WSLU vs. WDLS policies. (a) Human players. (b) LLM players.

**Setup** To distinguish between players adopting a purely random strategy and those aligning with a Nash equilibrium strategy, we employ a modified payoff matrix adopted by prior work. In this design, the Nash equilibrium corresponds to a mixed strategy of $\frac{1}{4}$ for Rock, $\frac{1}{2}$ for Paper, and $\frac{1}{4}$ for Scissors, while the uniform random strategy assigns equal probability $\frac{1}{3}$ to each action (Table 1(a)).

To define decision patterns based on the previous round's choice, we classify each transition into three types: stay (repeat the previous action; e.g., Rock $\rightarrow$ Rock), upgrade (choose the action that beats the previous one; e.g., Rock $\rightarrow$ Paper), and downgrade (choose the action that loses to the previous one; e.g., Rock $\rightarrow$ Scissors).

Firstly, we evaluated each of the six LLMs in a fully crossed Rock–Paper–Scissors tournament against all five other models and itself, with each pairing playing 50 rounds. The primary focus was to identify whether models exhibit frequency bias, sequential dependencies.

Secondly, we assessed how LLMs adapt when facing predictable, rule-based opponents by introducing two bots: one following a Win-Stay/Lose-Upgrade (WSLU) pattern and another following a Win-Downgrade/Lose-Stay (WDLS) pattern. This setup allows us to probe whether LLMs can detect structured opponent behavior and adjust their responses accordingly. For full details of match pairings, repetition counts, and transition matrices, see the appendix.

**Results** For the first stage of the experiment, we examine the proportion of choices made by each model to assess whether their behavior aligns with the theoretical Nash equilibrium or follows a purely random strategy. The choice distribution plots are shown as Figure 1.

In the second stage of our experiment that pairs LLMs with WSLU and WDLS bots, we summarize their win counts and cumulative payoffs in Table 3, and visualize their strategy distribution under each condition in Figure 2.

| Model | Elo Score |
|---|---|
| Claude-3.7 | 1670.95 |
| DeepSeek-R1 | 1601.81 |
| o1 | 1536.62 |
| Claude-3.5 | 1464.69 |
| DeepSeek-V3 | 1399.88 |
| GPT-4o | 1326.04 |

Table 2: Elo rankings for LLM–LLM matches in the RPS experiment.

| Strategy | Subject | Win Diff. | Payoff Diff. |
|---|---|---|---|
| WSLU | Humans | +2.01 | +6.30 |
| | LLMs | +4.50 | +14.56 |
| WDLS | Humans | +6.42 | +24.72 |
| | LLMs | −6.44 | −15.56 |

Table 3: Average win and payoff differentials (Subject − Bot) under WSLU vs. WDLS policies. Positive values mean the subject outperformed the bot.

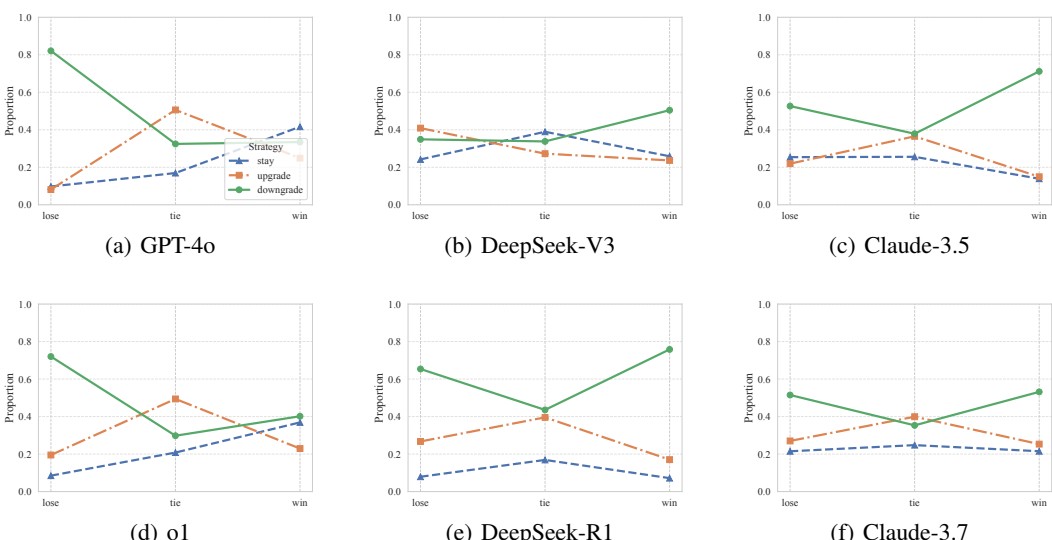

Figure 3: Outcome-conditioned strategy distributions, grouped by model class: (a) GPT-4o, (b) DeepSeek-V3, (c) Claude-3.5, (d) o1, (e) DeepSeek-R1, (f) Claude-3.7.

To probe model-level decision patterns, we analyze two complementary settings. First, in LLM–LLM matches we (i) compute each model's outcome-conditioned transition frequencies (Figure 3), and (ii) derive their head-to-head Elo rankings, reported in Table 2. Second, in LLM–bot trials we record every model's win-count differential and cumulative payoff differential against the WSLU and WDLS bots (Table 4). Together, these results reveal the qualitative heuristics favored by each LLM and their quantitative strength relative to both competing LLMs and rule-based opponents.

## 4.2 PRISONER'S DILEMMA

The game of Prisoner's Dilemma is a classic example of a social dilemma in which two players must simultaneously choose between two options: to cooperate or to defect. In theory, the optimal strategy under finite rounds is to defect, as defection strictly dominates cooperation in each individual round. However, empirical studies involving human participants have consistently shown that cooperation is not only possible but also fairly common, even in one-shot or finite-round versions of the game. (Ahn et al., 2001; Schneider & Shields, 2022)

One key factor that may help explain the gap between theoretical predictions and observed human behavior is the possibility of future interaction. Game theorists have long acknowledged that repeated play and the expectation of continued engagement can profoundly influence humans' strategic choices. When individuals anticipate future encounters, they often develop implicit systems of punishment and reward, which help deter opportunistic behavior and foster cooperation (Bó, 2005).

Building on this foundation, it is worthwhile to explore how LLMs behave under similar conditions in the Prisoner's Dilemma. By simulating strategic environments modeled after human studies (Bó,

| Model | WSLU | | WDLS | |
|---|---|---|---|---|
| | **Win** | **Payoff** | **Win** | **Payoff** |
| GPT-4o | **12.33** | **36.00** | −6.33 | −23.33 |
| DeepSeek-V3 | −3.67 | −8.67 | −5.67 | −15.33 |
| Claude-3.5 | 0.00 | 2.67 | −1.00 | −7.33 |
| O1 | 5.33 | 24.00 | −6.00 | −8.00 |
| DeepSeek-R1 | 7.00 | 15.33 | −20.33 | −52.67 |
| Claude-3.7 | 6.00 | 18.00 | **0.67** | **13.33** |

Table 4: Average win and payoff differentials by model under WSLU and WDLS conditions. Best values are **bold**; worst values are underlined.

| (a) Human study | | | | | (b) LLM experiments | | | |
|---|---|---|---|---|---|---|---|---|
| Dice | | Finite | | | Dice | | Finite | |
| **Treatment** | **Coop. (%)** | **Treatment** | **Coop. (%)** | | **Treatment** | **Coop. (%)** | **Treatment** | **Coop. (%)** |
| $\delta = 0$ | 9.17 | $H = 1$ | 10.34 | | $\delta = 0$ | 33.33 | $H = 1$ | 16.15 |
| $\delta = 0.5$ | 27.41 | $H = 2$ | 10.11 | | $\delta = 0.5$ | 38.37 | $H = 2$ | 21.35 |
| $\delta = 0.75$ | 37.64 | $H = 4$ | 21.43 | | $\delta = 0.75$ | 37.83 | $H = 4$ | 23.96 |

Table 5: Percentage of cooperation by treatment in (a) human study and (b) LLM experiments. Dice mode means that after each round the game continues with probability $\delta$, and Finite mode means that the game lasts for a fixed number of rounds $H$.

2005), we aim to examine whether and how LLMs adjust their behavior in response to the potential for future interactions.

**Setup**    The design of the LLM experiment follows that of the human study, with adjustments to accommodate the nature of AI agents. We employ two session structures in this experiment: *Dice* sessions inform the model that after each round the game continues with probability $\delta$, while *Finite* sessions specify a fixed horizon $H$ in the prompt. Each PD session runs all three treatments concurrently: Dice with continuation probabilities $\delta \in \{0, 0.5, 0.75\}$ and Finite with fixed horizons $H \in \{1, 2, 4\}$, where $H = 1/(1 - \delta)$. By matching the expected number of rounds across Dice and Finite treatments, we ensure a fair comparison of LLM decision patterns under equivalent temporal incentives. Dice sessions embed an explicit "shadow of the future," with $\delta$ controlling the strength of expected continuation, while Finite sessions present a known, fixed horizon without uncertainty. This contrast isolates the impact of future expectations on model cooperation and defection behavior. For full details of continuation schedules, pairing algorithms, and prompt templates, see the appendix.

**Results**    The primary outcome measure in our study is the *cooperation rate* per round, defined as the proportion of cooperative choices made by players in each round. Tables 4(a) and 4(b) summarize the average cooperation rates observed in the human and LLM experiments. We compute, for each LLM its overall cooperation rate in every experiment condition as shown in Table 6.

## 4.3    ANALYSIS

After collecting the behavioral data from the RPS and PD experiments, building on the metrics in human study, we first contrast model outputs with human benchmarks to probe bounded rationality in LLMs. We then drill down to model–family differences to uncover how model architecture, alignment and training paradigm shape models' strategic tendencies. The analysis is organized around two research questions.

RQ1: DO LLMS REPLICATE HUMAN PATTERNS OF BOUNDED RATIONALITY?

**LLMs mirror but amplify human heuristics.**    Across both games, models show similar qualitative departures from Nash equilibrium that typify human play, yet the magnitude of those deviations

| Type | Treatment | GPT-4o | DeepSeek-V3 | Claude-3.5 | o1 | DeepSeek-R1 | Claude-3.7 |
|---|---|---|---|---|---|---|---|
| Dice | $\delta = 0$ | 31.2 | 3.1 | 100.0 | 3.1 | 0.0 | 62.5 |
| | $\delta = 0.5$ | 48.4 | 22.7 | 100.0 | 9.0 | 1.6 | 65.2 |
| | $\delta = 0.75$ | 36.6 | 13.5 | 98.7 | 4.3 | 0.8 | 65.9 |
| Finite | $H = 1$ | 21.9 | 12.5 | 37.5 | 0.0 | 0.0 | 25.0 |
| | $H = 2$ | 37.5 | 25.0 | 25.0 | 3.1 | 0.0 | 37.5 |
| | $H = 4$ | 46.9 | 21.1 | 30.5 | 3.1 | 1.6 | 40.6 |

Table 6: Percentage of cooperation by LLM model, experiment type, and treatment.

is larger. In RPS, both humans and LLMs select Rock, Paper, and Scissors with near-equal frequencies instead of the calculated Nash equilibrium (Figure 1). However, LLMs show more concentrated choice distributions, while human participants show skewed preferences more often.

In PD, both humans and models increase cooperation under infinite-horizon treatments, consistent with the "shadow of the future" effect (Tables 4(a) and 4(b)). Yet, LLMs display an exaggerated sensitivity: even when $\delta = 0$, a condition theoretically equivalent to the one-shot game ($H = 1$), models show notably higher cooperation than in the finite counterpart. It shows LLMs may be overly influenced by the presence of future-oriented phrasing in the prompts, generalizing cooperative heuristics beyond the structural incentives observed in humans.

**LLMs show weaker environmental sensitivity.** Humans flexibly adapt heuristics to opponent structure, boosting win–stay by over 20 percentage points when facing the downgrade-oriented WDLS bot (Figure 2(a)). LLMs show only minimal adjustment with their dominant lose-downgrade behavior remaining unchanged (Figure 2(b)). This corresponds to humans outperforming both WSLU and WDLS bots, whereas LLMs beat WSLU by +4.50 wins but lose to WDLS by –6.44 wins (Table 3). A parallel pattern appears in PD: humans raise cooperation from 27.4% when $\delta = 0.5$ to 37.6% when $\delta = 0.75$, whereas LLMs actually dip slightly from 38.4% to 37.8% (Tables 4(a) and 4(b)), showing that models respond to the presence of a future but not to increases in continuation probability.

**Implications** The findings show that while LLMs adopt core bounded-rational heuristics found in humans, such as outcome-driven adjustments and future-driven cooperation, they do so in a more rigid, exaggerated form. Models exhibit fixed, stereotyped biases (e.g. lose–downgrade and baseline cooperation) instead of the heterogeneous mix of strategies seen in human players, and they are markedly less responsive to environmental cues like opponent rules or varying continuation probabilities.

RQ2: HOW DOES MODEL ARCHITECTURE INFLUENCE STRATEGIC BEHAVIOR?

**Model families exhibit distinct strategic signatures.** Outcome-conditioned transitions in LLM-LLM RPS games (Figure 3) reveal distinct architectural signatures: GPT-4o and o1 downgrade after a loss in more than 70% of cases, while Claude-3.5 and Claude-3.7 display a win-downgrade loop, preferring to downgrade even after victories. In PD (Table 6), only the Claude family sustains consistently high cooperation, whereas all other models remain below the 50% mark.

**Reasoning models excel under explicit equilibrium conditions.** The Elo ladder for LLM–LLM RPS play places all three reasoning models, Claude-3.7, DeepSeek-R1 and o1, in the top three slots (Table 2), confirming their advantages in such settings. Consistent with this, in PD, DeepSeek-R1 and o1 defect almost every round, exactly matching the Nash prediction, whereas the general-purpose models still cooperate a non-trivial fraction of the time (Table 6).

Figure 4 illustrates the contrast in round 1 of the $H=4$ treatment: Deepseek-R1 immediately defects and secures the highest possible payoff, while Claude-3.5 attempts to invite cooperation. The episode shows how reasoning models translate explicit payoff dominance into near-perfect play, whereas non-reasoning models balance exploitation and cooperation even when defection is analytically optimal.

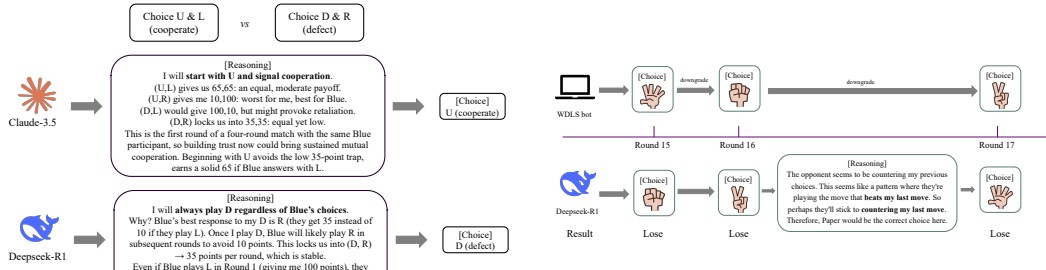

Figure 4: A case of Prisoner's Dilemma: round 1 in the $H{=}4$ treatment.

Figure 5: A case of Rock-Paper-Scissors: DeepSeek-R1 versus the WDLS bot, round 17.

**Reasoning models falter when opponent modeling is required.** When facing rule-based opponents, reasoning models show little advantage, and sometimes perform even worse than general LLMs: when against the WSLU bot they post only modest gains, and when against the WDLS bot DeepSeek-R1 records the worst result of all, with a win-count differential of $-20.33$ (Table 4).

Figure 5 records a critical slip in round 17 of the match: after noting the failure in round 16, R1 assumes that the bot will "counter my last move" and therefore downgrades to paper. This inference inverts the causal order, because the bot had already selected its round-16 action before it could observe R1's choice, the predicted stone response could never occur. The downgrade is punished at once by the bot's win-downgrade loop, triggering a sequence of losses that produces the large negative differential.

The case underlines a key weakness: when success hinges on theory-of-mind and causal reasoning about the opponent's information set, current reasoning models misapply their deductive heuristics and are outperformed by general LLMs.

**Implications** LLMs' strategic behavior is tightly bound to architectural lineage. Family-level biases indicate that pre-training and alignment choices hard-wire distinct decision styles. Reasoning models achieve textbook equilibrium when the solution is analytically explicit but fail to leverage theory-of-mind in rule-based adversarial settings. This gap highlights the importance of enhancing reasoning pipelines with richer opponent modeling and explicit theory-of-mind scaffolds so that strategic adaptation becomes as flexible as human play, rather than limited to problems with closed-form solutions.

## 5 CONCLUSION

We use protocols for human studies of Rock–Paper–Scissors (zero-sum) and the Prisoner's Dilemma (non-zero-sum) into text-based trials that place six leading LLMs under exactly the same payoffs, instructions and evaluation metrics used with humans. By replaying full sociological procedures rather than isolated prompts, our framework enables the first direct, fine-grained comparison of human and machine strategic behavior.

According to our analysis, while LLMs share key bounded-rational heuristics with humans, their rigidity and reduced sensitivity to context change mean they display only a partial, amplified form of human-like bounded rationality. Model-level analysis reveals two further insights: (1) each model family carries a distinct strategic fingerprint; and (2) reasoning models outperform general models when the optimal move is analytically obvious, yet struggle to infer and adapt to opponents' strategies, highlighting a persistent gap in theory-of-mind–driven reasoning.

In the future, we hope to integrate opponent-aware fine-tuning objectives, reinforcement learning from diverse human gameplay traces, and multi-stage prompts that explicitly invoke theory-of-mind, so as to equip LLMs with the flexible, context-sensitive strategic reasoning that characterizes human decision makers.

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

# A APPENDIX

## A.1 LLM DATA COLLECTION

We evaluate six state-of-the-art LLMs via their official APIs using identical prompts. The models and their names specified in APIs are as follows: GPT-4o (GPT-4o-2024-11-20), o1 (o1-2024-12-17), DeepSeek-V3 (deepseek-chat), DeepSeek-R1 (deepseek-reasoner), Claude-3.5-Sonnet (claude-3-5-sonnet-20241022), and Claude-3.7-Sonnet (claude-3-7-sonnet-20250219, with extended thinking).

To preserve behavioral variability and allow for probabilistic action distributions, we set the decoding temperature to 1.0 for all models. This high-temperature setting ensures that the models do not collapse to deterministic outputs and can express stochastic strategies when appropriate.

## A.2 ROCK-PAPER-SCISSORS

**Setup** In the first part, we examine how LLMs behave in an environment with minimal structure and no explicit opponent modeling. Each LLM plays repeated Rock-Paper-Scissors games for 50 rounds against six types of opponents: itself and each of the other five models. Every pairing is repeated three times to ensure robustness. This setup results in $\binom{6}{2} \times 3 = 45$ distinct matches, forming a fully crossed interaction network. To further quantify relative performance, we compute Elo scores for all models. Each model starts from a baseline rating of 1500, and ratings are iteratively updated after every match outcome using the standard Elo update formula.

The second part builds on prior human-subject studies that investigate how players adapt when facing structured, rule-based opponents. We use two bot opponents: one following a *Win-Stay/Lose-Upgrade* (WSLU) strategy, and the other following a *Win-Downgrade/Lose-Stay* (WDLS) strategy. Both bots operate probabilistically, with an 80% probability for their primary behavior after each outcome and 10% for each alternative action. In tie situations, actions are selected uniformly at random among the three options: stay, upgrade, and downgrade. The full transition probabilities for each strategy are presented in Table 7.

Each LLM is paired with both WSLU and WDLS bots for 50 rounds, repeated three times.

| Algorithm | Outcome | Stay | Upgrade | Downgrade |
|-----------|---------|------|---------|-----------|
| WSLU | Win | 0.80 | 0.10 | 0.10 |
|  | Tie | 0.33 | 0.33 | 0.33 |
|  | Lose | 0.10 | 0.80 | 0.10 |
| WDLS | Win | 0.10 | 0.10 | 0.80 |
|  | Tie | 0.33 | 0.33 | 0.33 |
|  | Lose | 0.80 | 0.10 | 0.10 |

Table 7: Probabilistic transition strategies used by the two bot opponents in the second experiment. Each row shows the probability of choosing *Stay*, *Upgrade*, or *Downgrade* following a given outcome.

**Prompts**  All prompts (exemplified) employed in this experiment are shown as below.

---

[**System Message**]
Rock-Paper-Scissors
You have been randomly paired with a computer algorithm (i.e., opponent) to play the Rock-Paper-Scissors game. You and your opponent will make decisions at the same time across multiple trials. In each trial, each of you will have to simultaneously select one of three options: Rock, Paper, or Scissors. The outcome of a trial (Lose, Win, or Tie) will depend on the decisions that both you and your opponent make according to these basic rules of the game:
– Rock beats Scissors
– Scissors beats Paper
– Paper beats Rock
Your payoff in a given trial will depend on the decisions that both you and your opponent make. **Beating the opponent brings you more points than tying (choosing the same option as the opponent), which brings you more points than if you get beaten by your opponent**. The exact payoffs for all possible outcomes are fixed throughout the game. After each trial, information will be provided to both you and your opponent about what both of you did and what were the corresponding payoffs in the previous trial.
Your payment will be given according to **the sum of points that you accumulate across all trials, as advertised, plus a base payment for successful completion of the study.**

[**Decision Message**]
Trial 6
Please make a choice: Rock, Paper, or Scissors. Think before you make your choice.
Output with the following format:
Reason: [Your reason for the choice]
Choice: [Rock/Paper/Scissors]

[**Feedback Message**]
Feedback in the previous trial:
You lost!
You chose Paper, your opponent chose Scissors.
Your payoff was 1, your opponent's payoff was 3.
Your total payoff so far is 42. Your opponent's total payoff so far is 42.

---

## A.3 PRISONER'S DILEMMA

**Setup**  We consider two types of session structures as below.

Dice: The model is informed that after each round, the game may continue with a fixed probability $\delta \in \{0, 0.5, 0.75\}$. The game terminates according to a simulated draw from a Bernoulli process. The continuation probability is made explicit to the model in its prompt.

Finite: The total number of rounds is known in advance and specified in the model's prompt. We use $H \in \{1, 2, 4\}$, matching the expected lengths of the infinite sessions. This enables a controlled comparison of decision patterns between sessions with and without a 'shadow of the future'.

To control for learning effects from one treatment to another, two sessions are run for each kind of continuation rule (Dice and Finite). For example, for Dice we run one session with the order ($\delta = 0$, $\delta = 0.5$, $\delta = 0.75$) and another with the inverse order ($\delta = 0.75$, $\delta = 0.5$, $\delta = 0$). We call the first kind of session "Normal" and the last kind "USD" (up-side-down).

A rotation matching scheme is used to avoid potential interaction and contagion effects between different repeated games. In each session, subjects are divided into two groups: *Red* and *Blue*. In each match, every Red subject is paired with a distinct Blue subject.

Given that each subject is matched only once with each subject from the opposite group, the total number of matches in a session is $\frac{N}{2}$, where $N$ is the number of subjects in a session. Since there are three treatments per session, each treatment contains $\frac{N}{6}$ matches.

For experiments conducted on LLMs, we set $N = 24$, with each LLM type participating as 4 distinct agents in a game. This configuration allows for up to 4 matches per treatment per subject, ensuring sufficient coverage and interaction diversity across treatments.

**Prompts** All prompts (exemplified) employed in this experiment are shown as below.

---

[**System Message**]
You are an undergraduate recruited for a research project.
The session you are participating in is broken down into 3 separate parts. At the end of the last part, you will be paid the total amount you have accumulated during the course of the 3 parts in addition to the show-up fee.
Your dollar earnings at the end of the session are determined by the points/$ exchange rate posted on the board in the front and back of the room. This exchange rate is equal to 200 points/$. Therefore, 200 points are equivalent to $1.
The participants are divided in two groups: Red and Blue.
Red and Blue participants will be matched together to interact in the following way. The Red participant can choose between U or D and the Blue participant can choose between L and R.
If the Red participant chooses U and the Blue participant chooses L, both earn 65 points.
If the Red participant chooses U and the Blue participant chooses R, the Red participant earns 10 and the Blue participant earns 100 points.
If the Red participant chooses D and the Blue participant chooses L, the Red participant earns 100 and the Blue participant earns 10 points.
If the Red participant chooses D and the Blue participant chooses R, both earn 35 points.
The points of the Red participants are indicated on the screen in red, and the Blue participant points are indicated in blue.
We will show the result of previous rounds of the current match.
Remember that you are a Red participant.

[**Dice-mode introduction**]
We will begin the first part now. This first part will consist of 4 matches. In each match every Red participant is paired with a Blue participant. You will not be paired twice with the same participant during the session or with a participant that was paired with someone that was paired with you or with someone that was paired with someone that was paired with someone that was paired with you, and so on. Thus, the pairing is done in such a way that the decisions you make in one match cannot affect the decisions of the participants you will be paired with in later matches or later parts of the session.
In this part, after each round we will roll a four sided dice. If the numbers 1, 2 or 3 appear, the participants will interact again without changing pairs. If a 4 appears, the match ends and participants are re-matched to interact with other participants. Therefore, in this part, each pair will interact until a 4 appears. After that, a new match will start with different pairs. Therefore you will interact until a 4 appears, with 4 different participants.
You will now participate in 4 matches, each match paired with a different participant. In each

---

match you will interact with the same person until a 4 appears. Remember: your decisions in one match cannot affect the decisions of the people you will interact with in future matches. This is not a practice; you will be paid!

If you are a Red participant you can choose the actions in red, U or D, and if you are a Blue participant you can press the actions in Blue, L or R.

[**Finite-mode introduction**]

We will begin the first part now. This part will consist of 4 matches. In each match every Red participant is paired with a Blue participant. You will not be paired twice with the same participant during the session or with a participant that was paired with someone that was paired with you or with someone that was paired with someone that was paired with someone that was paired with you, and so on. Thus, the pairing is done in such a way that the decisions you make in one match cannot affect the decisions of the participants you will be paired with in later matches.

In this part, each pair will interact once. You will now participate in 4 matches, each match paired with a different participant. In each match you will interact with the same participant once. Remember: your decisions in one match cannot affect the decisions of the people you will interact with in future matches. This is not a practice; you will be paid!

If you are a Red participant you can press the actions in red, U or D, and if you are a Blue participant you can press the actions in Blue, L or R.

[**Choice Message (New match)**]

You are now matched with a new participant. You will interact with this participant until a 4 appears. Make your choices now.

Think before you make your choice.

Output with the following format:

Reason: Your reason for the choice

Choice: U/D

[**Feedback Message**]

Feedback in the previous rounds:

Your choices: U

Opponent choices: L

Your total payoff: 65

Opponent total payoff: 65

[**Dice-mode Continuation Message**] A 2 appeared therefore this match continues. Now you are in Round 2 of the same match. You are still interacting with the same participant. You can see in the history the result of the previous rounds. Make your choices now. Think before you make your choice.

Output with the following format:

Reason: Your reason for the choice

Choice: U/D

[**Dice-mode End Message**] A 4 appeared therefore this match ended. You have earned 65 points. Now you will be matched with the next participant.