# OpenReview forum: "Beyond Nash Equilibrium: Bounded Rationality of LLMs and humans in Strategic Decision-making"
_ICLR.cc/2026/Conference — ICLR 2026 Conference Withdrawn Submission_

### Official Review · Reviewer_D8AS · 2025-10-21

**Soundness:** 2
**Presentation:** 3
**Contribution:** 1
**Rating:** 2
**Confidence:** 3

**Summary:**

The paper investigates the bounded rationality of LLMs vs humans in strategic decision-making tasks. Using two canonical game-theoretic paradigms, Rock-Paper-Scissors (RPS) and Prisoner’s Dilemma (PD), the authors compare six major LLMs (GPT-4o, o1, Claude-3.5/3.7, DeepSeek-V3/R1) against documented human behavioural data. The study examines whether LLMs reproduce human-like deviations from Nash equilibrium under repeated interactions. The results show that while models display similar qualitative heuristics to humans, they do so in a more rigid and less adaptive manner, revealing limited sensitivity to opponent structure or environmental variation.

**Strengths:**

- Directly targets bounded rationality in simple, well-studied games with closed-form Nash benchmarks.
- Multiple model families, including reasoning-tuned variants were tested and compared. The per-model “strategic signatures” are interesting.

**Weaknesses:**

- Prompt-only simulations. Despite claims of aligning with human protocols, this remains a prompt-only setup. There is no real incentive, no real-time human interaction, and no direct head-to-head human play.
- Insufficient linkage to prior work. Many closely related studies are cited, but there is no quantitative comparison with these works.
- Rigidity and weaker environmental sensitivity are likely due to the internal biases of the model, but this is not further investigated.
- WSLU/WDLS analysis is under-specified. The paper reports that LLMs beat WSLU but lose to WDLS overall and that DeepSeek-R1 is worst vs WDLS yet it doesn’t provide what the “correct” response policy against each bot should be and subsequently diagnose why the models deviate.
- Limited formal testing of significance/robustness across experiments.  No causal or ablation analysis tying behaviours to architecture, alignment, or prompting choices.

**Questions:**

1. On prior work: Many earlier studies compare LLMs and humans in similar game-theoretic settings. Can the authors provide quantitative comparisons or direct benchmarks against these existing results to clarify what new empirical insight this paper adds?
2. On model rigidity: The paper attributes LLMs’ rigidity and weaker environmental sensitivity to bounded rationality but does not explore possible causes. How might this be tested experimentally?
3. On WSLU/WDLS evaluation: What is the theoretically optimal or “correct” response policy against each of these bots, and how do the authors’ observed model behaviours deviate from that baseline? A more detailed analysis here could reveal whether the LLMs’ responses were driven by strong internal biases rather than adaptive reasoning.
4. On statistical robustness: Were any formal statistical tests conducted to confirm the significance and robustness of behavioural differences across models and treatments? If not, can the authors include such analyses or clarify why descriptive reporting was preferred?

Minor clarification: The text refers to Tables 4(a) and 4(b), but these appear to correspond to Tables 5(a) and 5(b). Could the authors confirm and correct this?

---

### Official Review · Reviewer_pL7m · 2025-10-25

**Soundness:** 1
**Presentation:** 1
**Contribution:** 1
**Rating:** 0
**Confidence:** 4

**Summary:**

The paper uses identical payoff structures and prompts to evaluate bounded rationality in LLMs versus humans by replaying human‐subject protocols in Rock-Paper-Scissors (RPS) and Prisoner’s Dilemma (PD). It reports LLMs are human-like but more rigid, with weaker environmental sensitivity, and family-specific strategic signatures, while reasoning models excel in clear cases but struggle with opponent patterns.

**Strengths:**

1. Evaluation across two complementary games (RPS and PD), covering both zero-sum and cooperative scenarios
2. Replicates human-subject protocols and payoff structures from behavioral economics literature

**Weaknesses:**

1. **Poor Organization.**
The presentation is disorganized because the authors put the methodology, setup, strategy description, results for both games, and overall analysis within the "Experiment" section. As a result, it is unclear where the experiment design ends and the analysis begins, forcing the reader to jump back and forth to follow the logic. The authors analyze RPS and PD together, even though they have different theoretical foundations, which should be discussed separately before synthesizing.

2. **Unknown Human Baseline.**
The paper repeatedly refers to "human" but never clarifies whether these results come from their own experiments or from previously published datasets. Evidently, these numbers are lifted from Zhang et al. [1] and Bó [2], not newly collected, but the authors do not state this.

3. **The authors often overstate and have inconsistent claims.**
- The authors state that "LLMs and humans show near-equal frequencies in RPS" (line 390), but no statistical validation, only proportion figures that visually differ in shape and trend. I can clearly see that LLMs' decision preference is "Paper > Rock > Scissors", while human is equally distributed. Moreover, they state that "models increase cooperation under infinite-horizon treatments" (line 393), but in Table 6, 7/12 results fail to exhibit consistent increases in cooperation with larger $\delta$ or $H$; some (e.g., o1 and DeepSeek-R1) maintain extremely low cooperation rates across all conditions. These inconsistencies indicate that the behavior is neither robust nor generalizable. Therefore, the conclusion that "LLMs adopt core bounded-rational heuristics found in humans" (line 409) is not rigorous because the empirical evidence is inconsistent and does not justify such a strong claim.
- They also claim that "model families exhibit distinct strategic signatures" (line 417), but the results in Table 6 directly contradict this assertion. For example, the OpenAI family (GPT-4o and o1) shows significantly different cooperation rates across both game types (same for DeepSeek family and Claude models in Dice type). Besides, the authors generalize that "all other models (except Claude) remain below the 50% mark" (line 421), which is meaningless (the numerical thresholds should be statistically justified or clearly defined). Overall, the evidence does not support any coherent "family signature".

4. **Potential methodological issue.**
Since the authors do not provide any code, I can only refer to the prompt template in Appendix A.2. The prompt for RPS contains only the current trial number (e.g., "Trial 6") and feedback from the previous round. Without a full interaction history, it is impossible to identify any consistent patterns or long-term opponent strategies. Therefore, the claim that "LLMs show weaker environmental sensitivity" (line 399) cannot be substantiated because the experimental design itself deprives models of detecting the patterns from enough environmental context.

[1] Rock-Paper-Scissors Play: Beyond the Win-Stay/Lose-Change Strategy. Zhang et al.

[2] Cooperation under the Shadow of the Future: Experimental Evidence from Infinitely Repeated Games. Pedro Dal Bó.

**Questions:**

1. Can you try other payoff matrices to evaluate the consistency and robustness of models in decision-making?
2. Do you provide the full game history to models? If not, how can "environmental sensitivity" be measured?
3. Are the human results newly collected or directly copied from previous studies?
4. Did you conduct any statistical significance tests to confirm that the differences are significant?
5. Will you release code and records to support reproducibility and verification?


Format issues or typos:
1. Tables 4(a) and 4(b) in lines 364 and 393/394 should be Tables 5(a) and 5(b). The authors mixed two tables.
2. Figures 4 and 5 are too small.

---

### Official Review · Reviewer_jBdZ · 2025-10-26

**Soundness:** 3
**Presentation:** 3
**Contribution:** 3
**Rating:** 6
**Confidence:** 3

**Summary:**

This paper explores the bounded rationality of large language models (LLMs) by comparing their strategic decision-making to human behavior in classic game-theory settings. Using Rock-Paper-Scissors and the Prisoner’s Dilemma, the authors examine deviations from Nash equilibrium strategies. The results show that while LLMs mirror human heuristics and biases, they do so with greater rigidity and are notably less adaptive to dynamic environments. The study concludes that current LLMs only partially replicate human-like strategic reasoning, underscoring the need for better opponent modeling and context sensitivity.

**Strengths:**

1. **Clear and Logical Writing:** The paper is well-written, with a coherent and easy-to-follow structure.
2. **Principled Experiment Design:** Replicating established human behavioral studies provides a robust basis for comparing LLM and human behavior.
3. **Insightful Findings:** The observation that LLMs amplify human heuristics with greater rigidity—and show weaker environmental sensitivity—is a meaningful and nuanced contribution beyond simple claims of rationality.

**Weaknesses:**

1. The paper notes “distinct strategic signatures” across models but does not explore the underlying causes of these differences.

**Questions:**

See above

---

### Official Review · Reviewer_GuQq · 2025-10-30

**Soundness:** 1
**Presentation:** 2
**Contribution:** 1
**Rating:** 2
**Confidence:** 5

**Summary:**

This paper investigates the strategic behavior of large language models in game-theoretic settings, specifically examining how LLMs deviate from Nash equilibrium predictions in repeated games and whether these deviations mirror human strategic regularities. The research focuses on empirical analysis of heuristic strategies, particularly exploring patterns like "win-lose-stay" behaviors and attempting to identify what the authors term "strategic signatures" across different models. The work aims to bridge behavioral game theory and LLM behavior analysis by comparing model performance to human benchmarks in strategic interactions, with particular attention to the prisoner's dilemma and interactions with various algorithmic opponents.

**Strengths:**

The paper addresses an important and timely research question at the intersection of behavioral game theory and large language model behavior, making it relevant to AI decision-making capabilities. The empirical focus on repeated games and heuristic strategies is well-motivated, given the extensive literature on human departures from equilibrium play, and the research has clear potential to contribute meaningfully to our understanding of how LLMs behave in strategic settings. The comparative approach of examining LLM behavior alongside human patterns offers valuable insights into whether artificial systems exhibit similar cognitive biases and strategic regularities that characterize human decision-making in interactive environments.

**Weaknesses:**

The paper suffers from several significant methodological and presentation issues that undermine its conclusions. Most critically, the analysis lacks proper statistical rigor, with claims about "strategic signatures" and systematic patterns supported only by qualitative evidence rather than formal statistical testing with appropriate null models. The operationalization and measurement of key concepts like "win-lose-stay" patterns are not clearly defined or transparently reported, making it difficult to evaluate the robustness of the findings. Additionally, the paper presents an internal inconsistency regarding LLM consistency with human behavior, particularly when facing the Win Downgrade Lose Stay bot, which contradicts the central thesis without adequate explanation. The reliance on LLM self-explanations as evidence of underlying mechanisms is problematic given the substantial literature showing these rationalizations are often unfaithful to actual generative processes. Finally, the paper appears to miss engagement with closely related work like “Do Large Language Models Learn Human-Like Strategic Preferences?” and suffers from clarity issues in writing and experimental description that hinder reproducibility and assessment.

**Questions:**

No questions.

---

### Note · Authors · 2025-11-12

I have read and agree with the venue's withdrawal policy on behalf of myself and my co-authors.